# Does the urban low-carbon transition promote the residents' health consumption?

**Pian Chen**[ORCID][1,2]*, **Yanchi Chen**[1]

1 Institute of Food and Strategic Reserves/Collaborative Innovation Center of Modern Grain Circulation and Safety, Nanjing University of Finance and Economics, Nanjing, Jiangsu, China, 2 School of Business, Yancheng Teachers University, Yancheng, Jiangsu, China

* cp818916@126.com

## Abstract

This work employs a fixed-effects model to analyze the influence of urban low-carbon transition on residents' health consumption, utilizing data from the China General Social Survey (CGSS) for the years 2018 and 2021. The model estimation value is 0.1349, significant at the 5% confidence range, demonstrating that urban low-carbon transition enhances residents' health consumption. This result successfully underwent both the robustness test and the endogeneity test. The mechanism test reveals that urban low-carbon transition influences residents' health consumption through two pathways: increasing health awareness and improving the level of habitat environment. The heterogeneity studies indicate that the impact of the urban low-carbon transition on residents' health consumption varies according to income levels, gender, and urban development stages. Higher-income people, female residents, and those in locations with advanced urban development significantly contribute to this influence. The study ultimately presents pertinent policy proposals aimed at enhancing the urban low-carbon transition, increasing the residents' understanding of health consumption, improving the level of habitat environment, and paying attention to coordinated and healthy development.

## 1. Introduction

As countries worldwide experience ongoing economic and social progress, there has been a tremendous improvement in people's living conditions. The global concern for people's health status has increased significantly, and there has been a notable improvement in awareness of health consumption. Consumer demand for health capital, also known as health consumption, has been steadily increasing [1]. The Global Wellness Institute reports that the global health economy was valued at $4.3 trillion in 2017 and increased to $5.6 trillion in 2023, reflecting an annual growth rate of 5%, surpassing the world economic growth rate of 4%. Per capita spending on healthcare has increased in several countries in recent years; however, health issues remain prevalent. The prevention of major infectious diseases has grown more crucial in the post-pandemic age, while chronic diseases have emerged as a significant global health concern [2]. Consequently, there is an increasing need for health consumption. Many countries have recommended expediting the establishment of an economic and social development model that promotes health in response to the increasing demand for health

**Data availability statement:** All relevant data are within the manuscript and its Supporting Information files.

**Funding:** The author(s) received no specific funding for this work.

**Competing interests:** The authors have declared that no competing interests exist.

consumption. This approach aims to foster the harmonious growth of health, economy, and society.

Urban regions serve as hubs of socio-economic activity. The OECD reports that metropolitan regions contribute around 70% of worldwide energy-related carbon emissions [3]. The conventional urban development paradigm is incompatible with the demands of a low-carbon economy, leading to the emergence of the urban low-carbon transition idea. The urban low-carbon transition primarily involves cities adopting low-carbon development by emphasizing the reduction of fossil energy consumption and greenhouse gas emissions, driven by technological and institutional innovations central to the development process [4–6]. Consequently, promoting urban low-carbon transformation and reducing carbon emissions is crucial for mitigating global climate change. Residents' health consumption, a crucial component of urban economic activity, influences both the quality of life of residents and the sustainable growth of cities. How does the development of urban low-carbon transitions influence residents' health-conscious consumption while achieving environmental objectives? Does it facilitate or impede residents' healthy consumption? Current research has not sufficiently addressed this question.

China, the most populous nation globally, has consistently dedicated itself to advancing the green and low-carbon transition. This work examines China as a case study, utilizing the 2018 and 2021 China General Social Survey (CGSS) information to investigate the mechanism by which urban low-carbon transition influences residents' health consumption behavior. This study seeks to offer practical guidelines for facilitating the low-carbon transition of global cities while enhancing the quality of life for the global population and effectively mitigating resource and environmental pressures worldwide. This paper's potential marginal additions include the fact that previous work predominantly examines the low-carbon transition from macroeconomic or environmental viewpoints, whereas emphasizing home health consumption offers a micro-level research perspective. The integration of micro and macro research methodologies enhances the knowledge of the intricacies and variety inherent in urban low-carbon transitions. Secondly, integrating personal health consumption with urban low-carbon transition introduces an interdisciplinary research dimension to the current literature. This facilitates the dismantling of disciplinary barriers and fosters collaboration and integration across many professions.

## 2. Literature review and theoretical analysis

### 2.1. Literature review

In the field of environmental economics, the study of the correlation between low-carbon development and consumption has consistently been a central topic. Consumption, one of the three key drivers of economic development known as "troikas," is widely regarded as the most promising means to stimulate long-term economic growth. However, it necessitates ongoing and careful monitoring.

Research at the macro level has mostly examined the effects of the low-carbon transition on consumption, particularly in relation to the transition to low-carbon energy and the consumption of clean energy [7–9]. Lee et al. [10] employed the difference-in-differences (DID) model and a spatial difference-in-differences (SDID) model to examine the impact of China's implementation of a low-carbon city pilot (LCCP) on the process of transitioning to cleaner energy sources. The findings demonstrated that the use of LCCP greatly accelerated the energy transition process. Drozdz et al. [11]examine the factors influencing decarbonization processes in urban and rural Poland, highlighting that coal remains the predominant source of electricity generation. They assert that low-carbon policymaking must prioritize

the decarbonization of coal usage and propose pertinent policy recommendations for the formulation of a sustainable energy strategy in Poland. Kime et al. [12] established an analytical framework to assess energy justice and equality, examining the effects of the low-carbon energy transition on Black, Indigenous, people of color, low-income, and other frontline groups, thereby promoting energy justice and equity. Zhao et al. [13] conducted a study on the green and low-carbon development of energy-intensive industrial parks in China. They developed a multi-objective optimization model and a comprehensive evaluation framework to assess the potential for coordinated reduction of carbon emissions and air pollutants. The research findings indicate that adjusting the energy structure is a key strategy for promoting green and low-carbon growth.

At a small scale, research on the effects of transitioning to a low-carbon economy on consumption primarily centers around the low-carbon and green consumption patterns of households and people [14–16]. Cauvain and Andrew [17] assert that social housing providers in Greater Manchester have become innovators in the UK's low-carbon transition, effectively correlating carbon emission reductions with enhancements in the quality of life for low-income residents. Schrage and Kristin [18] examine the effects of interventions in the urban low-carbon transition on low-carbon living consumption in Nordic cities, revealing that existing strategies predominantly depend on non-binding measures in mobility and housing, along with various forms of household autonomy. Zhang et al. [19] employed the multi-region input-output (MRIO) approach to examine the variations in carbon emissions among households with diverse economic statuses and socioeconomic attributes. The study shows that when the economic level increases, changes in household consumption structure and carbon intensity might result in a reduction in carbon emissions. In their study, Pang et al. [20] employed input-output tables to calculate the indirect carbon emissions from household consumption (ICEHC) of both urban and rural middle-income groups. The findings indicated that the ICEHC was increasing among middle-income groups, with consumption patterns playing a significant role in facilitating the achievement of the ICEHC within this demographic.

Current studies mostly center on the effects of transitioning to a low-carbon economy on energy use, household consumption, and related factors. Health consumption is not only a significant component of household consumption but also a crucial means of protecting individual health. Hence, it is crucial to thoroughly investigate the effects of the low-carbon transition on citizens' health consumption. However, there is a scarcity of comprehensive literature that thoroughly investigates this phenomenon. Prior research has overlooked the influence on the health consumption of people during the urban low-carbon transition. Furthermore, previous studies have overlooked the investigation of the process by which the urban low-carbon transition affects residents' health consumption. This study empirically examines the effects of the urban low-carbon transition on residents' health consumption and its underlying mechanisms. It utilizes two sets of data from the China General Social Survey (CGSS) in 2018 and 2021 to address the limited research in this area.

## 2.2. Theoretical analysis

This article posits that the urban low-carbon transition has the potential to impact residents' health consumption from both the supply side and the demand side.

**2.2.1. Supply-side analysis.** From the supply side, the quantity and quality of products supplied have an impact on market supply and demand. The supply and demand hypothesis posits that an increase in supply and quality can enhance demand, hence elevating overall market efficiency and welfare. The urban low-carbon transition aimed at enhancing residents' health consumption can be examined from the supply

side in the following dimensions. First, based on the theory of supply-induced demand, increasing the supply of healthy products is the key to promoting healthy consumption. The advancement of low-carbon transition, coupled with the modification of urban industrial structures and technical innovation, can significantly enhance the development and use of low-carbon products, offering consumers a greater selection of sustainable and health-conscious options. This allows inhabitants to select products and services that align more closely with a healthy lifestyle based on their requirements and interests. Examples include organic food, eco-friendly home equipment, and health and wellness services to address inhabitants' health requirements. Second, according to the notion of information asymmetry, enhancing the quality of health products and services is essential. China's health industry, still in its developmental phase, faces issues such as the absence of industry standards, inconsistent quality, and misleading advertising, which have, to some extent, undermined consumer confidence in health consumption and fostered the misconception that "health consumption is an IQ tax." During the urban low-carbon transformation, the Government has enhanced market transparency and product credibility by establishing and enforcing rigorous regulations and standards, as well as bolstering the certification and management of low-carbon products and services. This effectively addresses the issue of information asymmetry, safeguards consumer rights and interests, and reinforces residents' confidence and satisfaction in health consumption. Third, integrating public goods theory with externality theory, augmenting the provision of health infrastructure is a crucial method to enhance health consumption during urban low-carbon transitions. The urban low-carbon transition facilitates the development of urban infrastructure by enhancing green spaces and advocating for low-carbon transportation [21], so ensuring a fundamental guarantee for inhabitants' health consumption. The urban low-carbon transition will expand parks and urban green spaces, offer additional leisure and exercise facilities, and foster residents' healthy consumption, while promoting low-carbon transportation methods such as walking and cycling to further enhance healthy consumption [22].

**2.2.2. Demand-side analysis.** From the demand side, consumers' demand behaviors, preferences, purchasing power, and ability to access information together shape the supply and demand pattern of the market. The low-carbon transition of urban areas to encourage healthy consumption among inhabitants can be examined from the demand perspective in the following manners. First, consumer preference theory posits that consumer choice behavior arises from the interplay between individual preferences and the surrounding environment. As environmental awareness and health information dissemination improve, residents are increasingly prioritizing healthy and eco-friendly lives [23], resulting in a progressive enhancement of their health consciousness. The low-carbon transformation of urban areas offers healthier food options, ample green health services, and improved living conditions, thereby fulfilling inhabitants' desire for a healthy lifestyle and altering consumption patterns, establishing healthy consumption as a prevailing trend. Second, from the standpoint of income impact theory, urban low-carbon transitions frequently coincide with the robust growth of green businesses, generating new employment prospects and revenue sources for inhabitants [24]. The increase in citizens' income levels improves their capacity to acquire health items and services, establishing a robust economic basis for the growth of health consumption. Third, drawing on the principles of economies of scale and cost-effectiveness, governmental assistance for green and low-carbon sectors during the urban low-carbon transition has facilitated the growth of industrial scale and technological advancement. The decrease in production costs has led to a decline in health product prices, hence enhancing citizens' excitement for

health consumption. Fourth, from the standpoint of mitigating information asymmetry theory, during the urban low-carbon transition, the Government and social organizations have successfully enhanced residents' awareness of healthy lifestyles through a series of educational initiatives focused on low-carbon practices, environmental protection, and health, thereby diminishing information asymmetry in healthy consumption. With increased awareness of healthy consumption, residents are more likely to select nutritious items in their purchasing selections, thereby fostering the ongoing expansion of healthy consumption. This leads to hypothesis 1.

Hypothesis 1: The urban low-carbon transition promotes the residents' health consumption.

**2.2.3. Analysis of mechanisms.** The shift to a low-carbon urban environment can help citizens become more mindful of their health, which is mostly shown in the following ways. First, low-carbon advertising can raise locals' consciousness of their health. With the use of low-carbon awareness campaigns and public service announcements, the government encourages citizens to adopt a healthy and environmentally friendly lifestyle. Locals' awareness of the inherent connection between their health and the urban low-carbon transition raises their sense of both environmental responsibility and health. Second, as the green health sector grows, locals' awareness of their health will rise. More health-related goods and services, green production, and pollution reduction are all necessary for this business to grow. Residents' awareness of their health has increased as a result of the proliferation of health facilities and the availability of more nutritious food on the market. People's knowledge of their health has steadily grown as the urban low-carbon transition progresses. In order to encourage citizens' consumption of healthful foods, greater emphasis will be paid to the demand for healthy products such as organic food, eco-friendly household goods, fitness centers, and health products. This leads to hypothesis 2.

Hypothesis 2: The urban low-carbon transition promotes residents' health consumption by improving residents' health awareness.

The low-carbon transition can facilitate the development of urban green space systems, including parks and green belts, which enhance the urban environment, offer recreational areas, and improve inhabitants' living conditions, so further encouraging healthy consumption. first, a good living environment effectively enhances residents' sense of well-being, satisfaction and security [25]. This fosters individuals' zeal for life and enhances their focus on health, hence augmenting health-related expenditures. Second, the extension of urban green spaces, enhancement of public leisure places, and incorporation of sports and fitness facilities increase residents' willingness to engage in physical exercise and recreational activities, hence fostering the rise of health-related consumption. Third, as the quality of the human environment enhances, a healthy and eco-friendly lifestyle has progressively emerged as a societal trend. Individuals exhibit their quest for an elevated quality of life through healthy consumption, so deriving greater satisfaction and subsequently encouraging such consumption throughout the populace. This leads to hypothesis 3.

Hypothesis 3: The urban low-carbon transition promotes residents' health consumption by improving the level of habitat environment.

## 3. Empirical design

### 3.1. Modelling

The benchmark model that follows is built in this research to investigate how the urban low-carbon transition affects residents' health consumption.

$$HC_{itj} = \alpha_1 + \alpha_2 TFP_{jt} + \alpha_3 CONTRIL_{ij} + \psi_t + \eta_j + \varepsilon_{ij} \tag{1}$$

In equation (1), $HC_{itj}$ denotes the residents' health consumption of city $j$ in year $t$. $TFP_{jt}$ indicates the low-carbon transition and development of city $j$ in year $t$. $CONTROL_{ij}$ represents a series of control variables, both at the individual level and at the city level. $\eta_j$ is the fixed effect of the city. $\psi_t$ is the fixed effect of the year. $\varepsilon_{ij}$ is the random error term. This paper clusters standard errors at the city level.

This paper draws on the research of Jiang [26] to examine the existence of two mechanisms: residents' health awareness and habitat environment level. The mediating effect model is constructed as follows:

$$M_{itj} = \kappa_1 + \kappa_2 TFP_{itj} + \kappa_3 CONTROL_{ij} + \psi_t + \eta_j + \varepsilon_{ij} \tag{2}$$

In equation (2), $M_{itj}$ is the mediating variable including residents' health awareness ( $HA$ ) and habitat environment level ( $EL$ ), and the other variables have the same meaning as equation (1).

### 3.2. Variable selection and measurement

Explained variable. The explained variable for this paper is residents' health consumption. Academics have defined "health consumption" from different perspectives, and this paper argues that health consumption refers to consumption activities for the purpose of maintaining, improving, or restoring one's health, which includes, but is not limited to, the purchase of medical services, health food, health insurance, etc., and can be categorized into active health consumption and passive health consumption. Active health consumption emphasizes positive actions and decisions by individual consumers, such as investing in health care insurance. Passive health consumption, on the other hand, relies more on external factors, such as the emergence of diseases. Compared with passive health consumption, active health consumption better reflects residents' willingness and ability to consume health. The purchase of commercial health insurance falls into the category of active health consumption, a behavior that reflects the high value residents place on health and their awareness of active management of future health risks, and is an important indicator of the level of health consumption. Therefore, this paper draws on the research of Mocan et al. [27] and uses the questionnaire of the CGSS to measure health consumption by "whether or not commercial health insurance is purchased". A value of 1 is assigned to "yes" and 0 to the rest.

Explanatory variable. This work focuses on the urban low-carbon transition as the explanatory variable. The urban carbon total factor productivity index is used to compute this variable, employing the super-efficient slacks-based measurement, data envelope analyses (SBM-DEA). The method used for calculating the index is based on the approach developed by Shao et al. [28] The computation involves measuring input components such as capital, labour, and energy, and output factors such as GDP and carbon emissions. Capital is quantified using the urban fixed capital stock, labour is quantified using the number of employees, energy is quantified using the conversion of liquefied petroleum gas to standard coal, natural gas, and electricity consumption of the entire society, and carbon emissions are quantified

using the urban carbon emissions data published by the China Emission Accounts & Data-sets (CEADs).

Mediating variables. The intermediate variables in this study are residents' health aware-ness and habitat environment level. When health issues affect residents' daily lives, residents pay more attention to their health. Therefore, residents' health awareness is measured by the question "How often have health problems affected your work or other daily activities in the past four weeks?" in the CGSS questionnaire. Habitat environment level is measured by Assaye et al. [29] as a logarithm of urban green space area.

Control variables. Referring to Yang and Zhang [30],Shen and Sun [31],Zhang et al.[32] and other studies, the level of urban development level ( $GDP$ ) and urban population size ( $PP$ )are selected as city-level control variables, and the population income ( $PI$ ), household $MS$ size ( $HZ$ ), education level( $EDU$ ), age( $AGE$ ),state of health( $HE$ ),the gender ( $GE$ ), marital status ( $MS$ ) are selected as individual-level control variables.

## 3.3. Data sources and descriptive statistics

Considering that consumer behavior is inherently individualistic and significantly shaped by internal views [33], altering consumer behavior patterns is a gradual endeavor. To advance sustainable development, the Chinese government has designated three cohorts of low-carbon pilot cities since 2010; the inaugural cohort comprised five provinces and eight municipalities, while by 2017, the third cohort included eight provinces, 80 municipalities, and one region recognized as low-carbon pilot cities. The execution of the low-carbon city pilot policy and the rise in the number of pilot cities have not only facilitated the consistent expansion of the market supply for low-carbon products but have also significantly expedited the pervasive propagation of the low-carbon concept across all societal sectors. The public's awareness of low-carbon health is being acknowledged and integrated into their daily consumption choices. Consequently, this analysis utilizes data from the 2018 and 2021 Chinese General Social Survey. The sample for this study comprises 15,831 samples collected from 56 cities. The China City Statistical Year-book, statistics yearbooks of prefectural-level cities, and China Emission Accounts & Datasets (CEADs) are sources of relevant data at the city level. In cases where data is absent, regional averages are used as replacements. Table 1 displays the descriptive statistics for the variables.

# 4. Analysis of empirical results

## 4.1. Analysis of baseline results

The findings from the benchmarking model analyzing the impact of the urban low-carbon transition on the residents' health consumption are displayed in Table 2. This article employs fixed city and fixed year effects in order to account for potential unobservable factors that may influence residents' health consumption. The regression test results are presented in Table 2. In Table 2, column (1) demonstrates the direct impact of urban low-carbon transition on res-idents' health consumption, and the regression results are significantly positive. Column (2) further examines the impact of urban low-carbon transition on residents' health consumption after adding control variables, and the regression result is still significantly positive. This sug-gests that the development of urban low-carbon transition significantly promotes residents' health consumption.

## 4.2. Robustness tests

In order to test the robustness of the baseline results, robustness tests were conducted by replacing explanatory variables, excluding part of the sample, and shrinking the tails, and the results are shown in Table 3. Given the positive effects of physical activity on health

**Table 1. Description of variables and descriptive statistics.**

| Variable | Definitions and Assignment | Mean | Sd |
|---|---|---|---|
| HC | In the CGSS: "whether or not commercial health insurance is purchased" to measure it. A value of 1 is assigned to "yes" and 0 to the rest. | 0.1223 | 0.3280 |
| TFP | Expressed in terms of urban total factor carbon productivity. | 0.3295 | 0.0996 |
| HA | In the CGSS: "In the past four weeks, how often has your work or other daily activities been affected by health problems" Always, often, sometimes = 1; other = 0. | 0.3004 | 0.4584 |
| EL | Expressed in logarithmic terms in terms of urban green space area. | 35.0370 | 9.5432 |
| GDP | Expressed in terms of urban GDP (in RMB trillions) | 0.6494 | 0.7417 |
| PP | Expressed as total year-end urban population (in millions) | 8.6476 | 8.0329 |
| PI | In the CGSS: What was your personal total income for the whole of last year (2017). (Unit: RMB 10,000) | 4.4665 | 26.6235 |
| HZ | In the CGSS: May I ask who else is in your family besides you. | 3.3465 | 1.8043 |
| EDU | In the CGSS: Your current highest level of education is: primary school, private school, literacy class, no education at all = 1; junior high school, vocational high school, general high school, secondary school, technical school = 2; university colleges (adult higher education), university colleges (formal higher education), university undergraduate (adult higher education), university undergraduate (formal higher education), postgraduate and above = 3. | 1.8314 | 0.7175 |
| AGE | In the CGSS: "What is your date of birth" age = 2024 - year of birth | 56.4443 | 17.2821 |
| HE | In the CGSS: Do you feel that your current physical health is: very unhealthy, relatively unhealthy = 1, average = 2, relatively healthy, very healthy = 3, don't know, refuse to answer = 0. | 2.3740 | 0.7860 |
| GE | In the CGSS: genders: male = 0, female = 1. | 0.4571 | 0.4981 |
| MS | In the CGSS: Your current marital status is: unmarried = 0, married = 1. | 0.7384 | 0.4395 |

promotion, those who frequently participate in physical activity usually show a stronger tendency to be health-conscious and are more likely to invest in sports and health-related consumption [34]. Using the CGSS questionnaire, "Whether or not to participate in physical activity during free time" to measure residents' health consumption, "every day" was assigned a value of 4, "several times a week" was assigned a value of 3, "several times a month" was assigned a value of 2, and "several times a year" was assigned a value of 1, and 'Other' was assigned a value of 0. The results are significant at the 10% confidence interval. Due to the restricted behavioral capacity of individuals aged 75 and older, this paper excludes samples beyond 75 years of age prior to replacing equation (1) for the analysis. The test findings are presented in column (2) of Table 3, and the estimates are significant at the 5% confidence level. Column (3) of Table 3 presents the results of substituting equation (1) following a 5% reduction in income level, age, urban development level, and urban population size. The results are significant at a 5% confidence level. The benchmark results successfully meet the aforementioned three robustness tests.

### 4.3. Endogenous issues

As the degree of low-carbon transformation in cities increases, the quality of the urban environment also improves accordingly. Improved environmental quality not only facilitates the supply of health products, but also effectively raises the health awareness of residents, thereby promoting increased health consumption. Conversely, an increase in the level of healthy consumption by residents will further enhance their demand for health product and urban environmental quality, which will in turn promote supply-side structural reform and prompt cities to accelerate the pace of low-carbon transformation. Therefore, there may be a bidirectional causal relationship between urban low-carbon transition and residents' health consumption.

**Table 2. Analysis of baseline results.**

| Variable | (1) | (2) |
|---|---|---|
| | *HC* | *HC* |
| *TFP* | 0.1651*** | 0.1349** |
| | (0.025) | (0.062) |
| *GDP* | | 0.1529** |
| | | (0.059) |
| *PP* | | -0.0120 |
| | | (0.014) |
| *PI* | | 0.0004* |
| | | (0.000) |
| *HZ* | | -0.0008 |
| | | (0.001) |
| *EDU* | | 0.0734*** |
| | | (0.007) |
| *AGE* | | -0.0019*** |
| | | (0.000) |
| *GE* | | 0.0078*** |
| | | (0.002) |
| *MS* | | -0.0118** |
| | | (0.005) |
| _cons | 0.0682*** | 0.0385 |
| | (0.008) | (0.057) |
| City fixed | YES | YES |
| Year fixed | YES | YES |
| N | 15831 | 15831 |
| $R^2$. | 0.0025 | 0.0911 |

Standard errors are in brackets, *p < 0.1; **p < 0.05; ***p < 0.01

**Table 3. Robustness test results.**

| Variable | (1) | (2) | (3) |
|---|---|---|---|
| | *HC* | *HC* | *HC* |
| *TFP* | 0.2243* | 0.1307** | 0.1350** |
| | (0.126) | (0.063) | (0.063) |
| Control variable | YES | YES | YES |
| City fixed | YES | YES | YES |
| Year fixed | YES | YES | YES |
| _cons | -0.0322 | 0.0191 | 0.0591 |
| | (0.104) | (0.053) | (0.057) |
| N | 15831 | 13366 | 15276 |
| $R^2$ | 0.1016 | 0.0915 | 0.1029 |

Standard errors are in brackets, *p < 0.1; **p < 0.05; ***p < 0.01

This article employs an instrumental variable method to mitigate the potential endogeneity issue arising from the probable reverse causation between urban low-carbon transition and residents' health consumption. Additionally, Fu et al. [35] utilize inverse temperature as an instrumental variable for estimate purposes. The inverse temperature phenomenon is contingent solely upon random meteorological conditions and is not directly associated with residents' health consumption, hence satisfying the exogeneity criterion of the instrumental variable. The phenomenon of inverse temperature inhibits convection between surface air and upper air, hindering the dispersion of pollutants from the ground. Consequently, cities that frequently experience inverse temperature tend to be more polluted, thereby intensifying their impetus to adopt low-carbon transition strategies, thus fulfilling the correlation requirement of instrumental variables. The data for the intensity of the inversion comes from the Remote Sensing of Air Temperature database on the NASA website, which constructs an indicator of the intensity of the inversion based on the temperature difference between the first and third layers near the surface. The estimation outcomes derived from the two-stage least squares method is presented in Table 4. The results of the first stage of the instrumental variable method show that as the intensity of the inversion rises in each region, the stronger the city promotes the development of the low-carbon transition, and it is significant at the 1% level. The F-value of the first stage is greater than 10, ruling out the weak instrumental variable problem. The results of the second stage show that after using the intensity of inversion temperature as an instrumental variable, the development of urban low carbon transition significantly contributes to the health consumption of residents and is significant at the 1% level. This indicates that the estimation results in this paper are robust after overcoming the endogeneity problem.

## 4.4. Mechanism analysis

### 4.4.1. Residents' health awareness.
This work utilizes the methods put forward by Jiang [26] to conduct mechanism testing. Table 5 displays the results. The regression results in column (1) of Table 5 demonstrate a significant relationship between urban low-carbon

**Table 4. Endogeneity test results.**

| Variable | (1) | (2) |
|---|---|---|
| | *TFP* | *HC* |
| *TFP* | | 0.5204*** |
| | | (0.141) |
| *IV* | 0.0004*** | |
| | (0.000) | |
| Control variable | YES | YES |
| _cons | 0.2609*** | -0.1189** |
| | (0.007) | (0.048) |
| City fixed | YES | YES |
| Year fixed | YES | YES |
| Weak instrumental variable test | 483.1795 | |
| | {16.38} | |
| N | 15831 | 15831 |
| $R^2$ | 0.0728 | 0.0650 |

The brackets in the weak instrumental variable test are the critical value of more than 10% in the Stock-Yogo test. Standard errors are in brackets, *p < 0.1; **p < 0.05; ***p < 0.01.

**Table 5. Mechanism test results.**

| Variable | (1) | (2) |
|---|---|---|
| | *HA* | *EL* |
| *TFP* | 0.0957* | 6.5557** |
| | (0.054) | (2.578) |
| Control variable | YES | YES |
| _cons | 0.8329*** | 25.4642*** |
| | (0.055) | (1.630) |
| City fixed | YES | YES |
| Year fixed | YES | YES |
| N | 15831 | 15831 |
| R² | 0.3589 | 0.9761 |

Standard errors are in brackets, *p < 0.1; **p < 0.05; ***p < 0.01.

transition and residents' health awareness. This significance is observed within a 10% confidence interval, showing that urban low-carbon transition has a substantial impact on enhancing residents' health awareness. Sheth et al. [36] introduced a theory of consumer value that posits that consumers make purchasing decisions based on the benefits they derive in five specific domains: social, emotional, functional, cognitive, and conditional. Multiple studies have demonstrated that personal values have a significant impact on consumer decision-making [37–39]. As health awareness rises, individuals are more inclined to acknowledge the importance of health behaviors, leading to an increase in health consumption habits. Thus, it may be deduced that the shift towards a low-carbon urban environment can enhance inhabitants' health consumption by enhancing their knowledge of health. Hypothesis 2 has been validated.

**4.4.2. Habitat environment level.** Column (2) of Table 5 presents the findings of the mediation analysis concerning habitat environment level. Column (2) of Table 5 demonstrates that the regression coefficient of the urban low-carbon transition on habitat environment level is significant at the 1% confidence level, showing a substantial enhancement in habitat environment level due to the urban low-carbon transition. According to Grossman's [1] health demand model, health is regarded as a type of human capital, and its demand is influenced by an individual's stock of health capital, the depreciation rate of health capital, and investment in health. Habitat improvements, such as improved air quality and increased greenery, can be considered as a health investment that can reduce the depreciation rate of health capital and increase investment in health and leisure consumption. Stanca and Veenhoven [40] also showed in their study that a good living environment can increase people's sense of well-being, which further enhances residents' health awareness and willingness to consume, thus promoting the growth of health consumption. Consequently, urban low-carbon transformation can enhance residents' health consumption by elevating the level of the habitat environment level.

## 5. Heterogeneity analysis

### 5.1. Heterogeneity in population income levels

This study aims to examine potential disparities in the effects of urban low-carbon transition on residents' health consumption across various income brackets. The average income of residents is used as the criterion for dividing the entire sample into two groups: high-income

and low-income. The analysis then reverts back to equation (1). The findings are displayed in Table 6. Columns (1) and (2) in Table 6 demonstrate that urban low-carbon transition positively affects the health consumption of high-income groups, whereas it does not have a substantial impact on the health consumption of low-income groups. This observation aligns with the results of the study conducted by Liu et al. [41]. The potential factors contributing to this issue are as follows: High-income groups typically possess greater financial capacity and better levels of education, resulting in a heightened concern for quality of life and environmental preservation. Consequently, they are more inclined to invest additional funds in eco-friendly health products and services. The potential factors contributing to this issue are outlined below. Furthermore, individuals belonging to low-income groups are more prone to experiencing heightened economic burdens, which may result in their concentration being primarily directed towards fulfilling essential necessities of life. The implementation of the low-carbon transition and development leads to the introduction of environmentally friendly products and facilities, which are frequently accompanied by higher costs. This might provide a challenge for low-income households in terms of affordability. Furthermore, individuals from low-income groups may face challenges in obtaining knowledge regarding health consumption and may encounter difficulties in adopting healthy lifestyles as a result of the limitations imposed by their living environment and working conditions.

## 5.2. Gender heterogeneity

This study aims to examine potential disparities in the effects of urban low-carbon transition on the health behaviors of inhabitants belonging to various genders. To do this, the entire sample is divided into separate groups based on gender, namely male and female groups. The findings are displayed in Table 6. Based on the data in columns (3) and (4) of Table 6, it is evident that the urban low-carbon transition has a substantial positive effect on the health consumption of female residents. However, it does not have a noticeable impact on the health consumption of male residents. The findings concur with those of Liu and Chen [42]. The potential causes are as follows: Firstly, women often assume the responsibility of nurturing and tending to the needs of a family. Consequently, women are more inclined to be persuaded and take action when cities actively encourage low-carbon, healthy lifestyles. This includes choosing healthier shopping choices, boosting household expenditure on health prevention,

**Table 6. Heterogeneity analysis.**

| | (1) | (2) | (3) | (4) | (5) | (6) |
|---|---|---|---|---|---|---|
| **Variable** | **Population income level** | | **Gender** | | **Urban development level** | |
| | *HC* | *HC* | *HC* | *HC* | *HC* | *HC* |
| *TFP* | 0.1907** | 0.0767 | 0.0520 | 0.0772** | 0.1120*** | 0.0479 |
| | (0.091) | (0.061) | (0.039) | (0.034) | (0.042) | (0.033) |
| Control variable | YES | YES | YES | YES | YES | YES |
| _cons | 0.0714 | 0.0647 | 0.0514 | -0.0152 | 0.0213 | 0.0156 |
| | (0.098) | (0.047) | (0.031) | (0.030) | (0.034) | (0.028) |
| City fixed | YES | YES | YES | YES | YES | YES |
| Year fixed | YES | YES | YES | YES | YES | YES |
| *N* | 7,764 | 8,067 | 7,237 | 8,594 | 7840 | 7991 |
| *R²* | 0.0838 | 0.0607 | 0.0878 | 0.0803 | 0.1056 | 0.0479 |

Standard errors are in brackets, *p < 0.1; **p < 0.05; ***p < 0.01.

and other related initiatives. Furthermore, men typically assume distinct roles and obligations within the family unit. Men may prioritize job advancement and economic wealth, among other factors, which could result in a relatively smaller influence of the low-carbon transition on their health consumption.

## 5.3. Heterogeneity in urban development level

This study examines the potential variations in the effects of urban low-carbon transition on the health consumption of inhabitants in cities with different degrees of development. The average urban GDP is employed as the criterion for dividing the cities. The complete sample is partitioned into two cohorts: one characterized by a higher degree of urban growth and the other characterized by a lesser degree. Subsequently, regression analysis is conducted for each group utilizing the formula (1). The precise findings are displayed in Table 6. Table 6 reveals that the low-carbon urban transition positively affects the residents' health consumption in the higher urban development group, but it does not have a significant impact on the health consumption of people in the lower urban development group, as evidenced by columns (5) and (6). This result aligns with the conclusions of Giles-Corti et al. [43]. The potential factors contributing to this issue are outlined below. In highly urbanized regions, residents tend to have higher disposable income, which enables them to allocate more funds towards healthcare expenditures. Regions characterized by minimal urban development may experience a dearth of economic resources, leading to limited access to healthcare services for residents. The full realization of the potential benefits of the low-carbon transition may be hindered by economic limitations. In contrast, regions characterized by greater urbanization exhibit superior infrastructure and a more extensive array of public services, hence fostering the uptake of healthcare. In regions characterized by minimal urbanization, the lack of proper infrastructure and public services might result in insufficient options and chances for locals to access and utilize healthcare resources.

## 6. Conclusions and policy implications

### 6.1. Conclusions

This research initially examines the theoretical process by which the urban low-carbon transition affects residents' health consumption. Furthermore, the fixed-effect model is employed to empirically examine the influence of urban low-carbon transition on residents' health consumption, utilizing the CGSS data from 2018 and 2021. First, the primary findings derived from the aforementioned study indicate that the urban low-carbon transition has produced a substantial impact on enhancing residents' health consumption. It is worth noting that this conclusion remains valid even after conducting a number of rigorous robustness tests and endogenous testing. Research has validated the influence of the urban low-carbon transition on citizens' health [44], although there is a paucity of studies that have specifically examined healthy consumption. Second, the shift towards a low-carbon urban environment might impact the way citizens consume health-related goods and services by enhancing their awareness of health and improving the level of habitat environment. Third, the effect of transitioning to a low-carbon urban environment on residents' health consumption varies according on their income level, gender diversity, and the amount of urban growth. The urban low-carbon transition has a considerable impact on the health consumption of residents in the high-income group, but not in the low-income group. Regarding gender disparities, the effect of transitioning to a low-carbon urban environment on residents' health consumption is noteworthy among females, but not among males. The influence of urban low-carbon transformation on residents' health consumption is large in areas with high levels of urban development, but not in areas with low levels of urban development.

## 6.2. Policy implications

First, enhancing the urban low-carbon transition. The findings indicate that the transition towards a low-carbon urban environment can enhance residents' health consumption and stimulate the growth of green and low-carbon economic and social development. The urban low-carbon transition will enhance the health consumption of Chinese inhabitants and contribute to China's objective of achieving "carbon peak and carbon neutrality." This fosters the establishment of a robust China and offers a consistent internal impetus for China's economic and social progress. Hence, nations across the globe can consolidate the accumulated knowledge of urban low-carbon transition to establish representative instances of low-carbon development and disseminate the successful practices globally.

Second, enhancing the residents' understanding of health consumption. Raising citizens' awareness of health consumption can be promoted through the urban low-carbon transition. Hence, it is imperative for the government of low-carbon transition cities to prioritize the dissemination and educational initiatives pertaining to low-carbon and healthy lives. This includes intensifying the promotion of environmentally-friendly habits such as "low-carbon consumption," "health consumption," and "green consumption." These activities can enhance people's understanding of the advantages of low-carbon transition, increasing residents' knowledge of health-conscious consumption and encouraging the adoption and development of healthy lifestyles.

Third, improving the level of habitat environment. The low-carbon transformation of cities can promote healthy consumption by improving the level of the human environment. Therefore, the governments of cities in low-carbon transition should strengthen the construction and improvement of green public infrastructure, and improve the construction of green areas, health theme parks and green health-care venues in cities. This can provide residents with places for leisure and exercise and increase health consumption.

Fourth, paying attention to coordinated and healthy development. The influence of urban low-carbon transition on residents' health consumption varies significantly among cities with different income categories and degrees of development. Hence, throughout the course of urban low-carbon transition, the primary focus should be on continuously enhancing the income level of all people. The government ought to augment financial support for individuals with low incomes and prioritize their healthcare requirements. The government should actively encourage the growth of environmentally-friendly industries, enhance job prospects, and enhance the purchasing power of individuals with low incomes. Furthermore, it is imperative to focus on the fundamental circumstances surrounding urban growth. It is important to carefully consider the effects of transitioning to a low-carbon economy on the residents' health consumption in cities with modest levels of development. The government should implement effective strategies to enhance the living standards and health consumption levels of urban people during the low-carbon transformation process.

## 6.3. Research limitations

Although this study provides new impetus for cities to further advance low-carbon transformation and development, and clarifies the impact of urban low-carbon transformation on residents' health consumption by considering health consumption awareness and habitat environmental level, it still has certain limitations. Due to constraints in the database, this article's analysis did not delve into specific aspects of medical and healthcare consumption. Future research will aim to explore in depth the impact of urban low-carbon transformation on residents' medical and healthcare expenditures.

## Supporting information

**S1 Data.** Data.
(XLSX)

## Acknowledgements

We would like to thank all members of the Doctoral Program in Collaborative Innovation Center of Modern Grain Circulation and Safety, and all support from the Nanjing University of Finance and Economics for making it possible to carry out this work.

## Author contributions

**Conceptualization:** Pian Chen, Yanchi Chen.

**Data curation:** Pian Chen, Yanchi Chen.

**Methodology:** Pian Chen.

**Project administration:** Pian Chen.

**Resources:** Pian Chen, Yanchi Chen.

**Writing – original draft:** Pian Chen, Yanchi Chen.

**Writing – review & editing:** Pian Chen.

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
