## [Decision Letter · Decision Letter 0]

1 Dec 2024

PONE-D-24-39039Does the urban low-carbon transition promote the residents' health consumption?PLOS ONE

Dear Dr. Chen,

Thank you for submitting your manuscript to PLOS ONE. After careful consideration, we feel that it has merit but does not fully meet PLOS ONE’s publication criteria as it currently stands. Therefore, we invite you to submit a revised version of the manuscript that addresses the points raised during the review process.

We look forward to receiving your revised manuscript.

Kind regards,

Najmeh Mozaffaree Pour

Guest Editor

PLOS ONE

Reviewers' comments:

Reviewer's Responses to Questions

**Comments to the Author**

1. Is the manuscript technically sound, and do the data support the conclusions?

Reviewer #1: Yes

Reviewer #2: Partly

Reviewer #3: Yes

2. Has the statistical analysis been performed appropriately and rigorously? 

Reviewer #1: Yes

Reviewer #2: Yes

Reviewer #3: Yes

3. Have the authors made all data underlying the findings in their manuscript fully available?

Reviewer #1: Yes

Reviewer #2: Yes

Reviewer #3: Yes

4. Is the manuscript presented in an intelligible fashion and written in standard English?

Reviewer #1: Yes

Reviewer #2: Yes

Reviewer #3: Yes

5. Review Comments to the Author

Reviewer #1: Environmental issues are always in the spotlight, and it was an interesting choice of topic to start with urban low carbon, and I started reading the manuscript with great interest. You have systematically argued and articulated the conclusions of this research very well. However, there are still some outstanding things that I hope to gain further enhancement through your improvement.

Abstract

1. The abstract follows a logical structure and provides a good overview of the study by summarizing the purpose, methodology, results, and conclusions. However, unfortunately, I did not see specific statistical results. Providing key results could have strengthened the impact of the study findings.

2. should have added the policy recommendations of the study, etc. after the conclusions of the study.

Presentation

1. line 40, progressive growth in health needs. Providing relevant data would better visualize this.

2. Lack of a clear definition of “urban low carbon”.

Literature Review and Theoretical Analysis

1. The literature review analyzes the current state of research in a comprehensive way, but your argument is mainly based on Chinese scholars and not on research in other countries.

2. In your theoretical analysis, it would be more comprehensive if you add more specific theories or formulaic derivations in the supply and demand analysis.

Analysis of mechanisms

For your explanatory variables, you have used the question “Are you currently enrolled in any of the following social security programs? Urban basic medical insurance/new rural cooperative medical insurance/public medical care cooperative medical insurance/public medical care; urban/rural basic pension insurance; commercial medical insurance; commercial pension insurance. insurance.” is measured by responses to a questionnaire. But this does not reflect the full picture of consumer spending, what is your rationale for measuring it this way?

Analysis of empirical results

In your section on heterogeneity analysis, you suggest comparing the findings of other scholars to increase credibility.

Conclusion and Policy Implications

The discussion section is too brief and does not go into depth from the findings. It should be integrated with the existing literature and suggest policy implications at the government level.

Reviewer #2: This paper uses fixed effects models to discuss the impact of low-carbon transformation on residents' health consumption and explores the mechanisms from the perspectives of residents' health awareness and urban environmental quality. Although the paper has conducted endogeneity analysis and a series of robustness tests, there are still some issues that need further explanation and revision. The specific opinions are as follows:

1. In terms of marginal contribution: Since there has already been research focusing on the impact of low-carbon transition on residents’ health or consumption, the authors should emphasize why it is important to focus on residents’ health consumption within the context of low-carbon transition research. The article should clarify what questions it addresses, what research perspective it adds to the existing literature, its practical significance, and its theoretical implications. Furthermore, the authors mention that this article establishes a theoretical analysis framework to examine the impact of low-carbon transition on residents’ health consumption. However, the theoretical analysis framework presented in the following sections is based on existing literature and does not offer significant theoretical contributions. Therefore, the authors need to reconsider whether the contributions in this section are appropriate.

2. The logic of the introduction is somewhat tenuous, especially in the second paragraph. The author first mentions the reasons for implementing a low-carbon transition in the second paragraph, then abruptly shifts the topic to the current economic situation and introduces the importance of consumption. It is difficult to emphasize why the low-carbon transition should prioritize the importance of residents' health.

3. The micro data of this paper comes from CGSS database in 2018 and 2021. As far as I know, CGSS data are collected in many periods, so why the author only uses the data of these two periods, such as the data of previous years can also be used? If not, it should be specified in the article.

4. In terms of variable definition, the article defines health consumption based on the purchase of four types of insurance. This method of defining variables is skeptical, and the author should clarify whether any existing literature has adopted this approach. Among the four types of insurance are health insurance and pension insurance; however, the relationship between pension insurance and residents’ health consumption is not closely related. Additionally, the author assigns values based on the number of insurance purchases, which fails to accurately reflect the differences in health consumption among residents. This is because basic health insurance is a social welfare benefit that residents are entitled to. If residents purchase commercial health insurance, their premium expenditures and level of concern for health are often higher, but the variable definition used in the article does not accurately capture this. Existing research on residents’ health consumption typically uses their expenditure on medical and health services as a research variable. If there are relevant variables in the data used in this article, it is recommended that the author include them in the robustness check section to enhance the reliability of the research conclusions.

5. In terms of model specification, both Model 1 and Model 2 should account for clustered robust standard errors at the city level to avoid bias in the baseline regression results due to the correlation among individuals within the same city.

6. Mechanism Analysis Section: (1) The definition of the mechanism variable “residents’ health awareness” in this article may not be entirely appropriate. The article is based on the questionnaire questions “In the past four weeks, how often has your work or other daily activities been affected by health problems? Always, often, sometimes = 1; other = 0” to define residents’ health awareness, a potential issue with the questionnaire is that the questions may reflect more on the extent to which residents are affected by health issues in their work and daily lives, rather than their concern for their own health problems. (2) Unlike existing literature that uses urban air quality as a representation of urban environmental quality, this article’s choice of urban green space area as a proxy variable for urban environmental quality may be incomplete. It is recommended that the authors appropriately include air quality for examination in this section.

7. Robustness Check Section: (1) Based on the research subjects and scenarios of this paper, using a fixed effects model for estimation is more appropriate. However, the use of a random effects model in this section as a robustness check does not resolve the issues present in the baseline estimation results; rather, the results obtained from the random effects model are coarser than those from the fixed effects model. Additionally, there is a problem in the presentation of the estimation results in this section; according to the estimation results in Table 3, Column 1, it should be a 90% confidence interval, not a 95% confidence interval. (2) The author should not only apply winsorization to income but should also apply it to all continuous variables in the control variables to mitigate the bias caused by outliers on the baseline regression results. (3) The author should consider adding more robustness checks, such as replacing the dependent or independent variables, or changing the level of fixed effects or so on, to enhance the credibility of the baseline regression results.

8. Endogeneity Analysis Section: To address potential endogeneity issues in this article, the author uses the urban green space coverage rate from 2006 as an instrumental variable for estimation. However, a possible problem is that this instrumental variable may not satisfy the exclusion restriction assumption, as the urban green space coverage rate can directly affect residents’ health rather than only indirectly influence residents’ health through its impact on urban low-carbon transformation. Therefore, it does not meet the exclusion condition. The author should include relevant tests for the exogeneity and exclusion restriction of the instrumental variable to assess its reliability.

Reviewer #3: 1. The contribution in the Introduction section is not sufficiently clear. Please reorganize this section and explicitly articulate the novelty of your paper in comparison to existing literature.

2. Please clarify whether calculating TFP at the city level can effectively capture industry heterogeneity within the city. Service and tourism industries may not experience the same degree of carbon reduction in their transformation.

3. Please specify whether the empirical results are based on clustered standard errors.

4. The F-statistic for the weak instrument is high, and the coefficient of IV is similar to OLS. Please discuss whether the instrument satisfies the exogeneity assumption.

6. PLOS authors have the option to publish the peer review history of their article (what does this mean? ). If published, this will include your full peer review and any attached files.

**Do you want your identity to be public for this peer review?** For information about this choice, including consent withdrawal, please see our Privacy Policy .

Reviewer #1: No

Reviewer #2: No

Reviewer #3: No

---

## [Author Response · Author response to Decision Letter 1]

12 Dec 2024

Reviewer #1

Abstract

Comments 1. The abstract follows a logical structure and provides a good overview of the study by summarizing the purpose, methodology, results, and conclusions. However, unfortunately, I did not see specific statistical results. Providing key results could have strengthened the impact of the study findings.

Response: Thanks for your useful suggestions. We have added key statistical results to the abstract as suggested by the reviewer.

Please see Lines 25-26 in the revised manuscript.

Comments 2. should have added the policy recommendations of the study, etc. after the conclusions of the study.

Response: Thanks for your useful suggestions. We have added relevant policy recommendations to the summary as suggested by the reviewer.

Please see Lines 34-38 in the revised manuscript.

Presentation

Comments 1. line 40, progressive growth in health needs. Providing relevant data would better visualize this.

Response: Thanks for your useful suggestions. We have added data related to the growth of global health consumption as suggested by the reviewer.

Please see Lines 45-48 in the revised manuscript.

Comments 2. Lack of a clear definition of “urban low carbon”.

Response: Thanks for your useful suggestions. We have added a definition of “urban low-carbon transition” as suggested by the reviewer.

Please see Lines 60-63 in the revised manuscript.

Literature Review and Theoretical Analysis

Comments 1. The literature review analyzes the current state of research in a comprehensive way, but your argument is mainly based on Chinese scholars and not on research in other countries.

Response: Thanks for your useful suggestions. We have added relevant studies by foreign scholars in the literature review section.

Please see Lines 100-108, 116-122 in the revised manuscript.

Comments 2. In your theoretical analysis, it would be more comprehensive if you add more specific theories or formulaic derivations in the supply and demand analysis.

Response: Thanks for your useful suggestions. We have added the relevant theory to the supply side and demand side analysis respectively.

Please see Lines 149-211 in the revised manuscript.

Analysis of mechanisms

Comments 1. For your explanatory variables, you have used the question “Are you currently enrolled in any of the following social security programs? Urban basic medical insurance/new rural cooperative medical insurance/public medical care cooperative medical insurance/public medical care; urban/rural basic pension insurance; commercial medical insurance; commercial pension insurance. insurance.” is measured by responses to a questionnaire. But this does not reflect the full picture of consumer spending, what is your rationale for measuring it this way?

Response: Scholars have delineated “health consumption” from various viewpoints, and this paper contends that health consumption encompasses activities aimed at maintaining, enhancing, or restoring health. This includes, but is not confined to, the acquisition of medical services, nutritious food, health insurance, etc., and can be classified into active and passive health consumption. Active health consumption highlights the proactive choices and behaviors of individual consumers, such as investing in health insurance. Conversely, passive health consumption is contingent upon external events, such as the onset of diseases. In contrast to passive health consumption, active health consumption more accurately represents people’ willingness and capacity to engage in health-related expenditures. Furthermore, while pension insurance does not directly offer health services or products, it indirectly promotes an individual’s health by guaranteeing financial stability. A stable income enables individuals to invest more liberally in their health, including access to healthcare services. This paper utilizes the research of Mocan et al. (2003) and integrates the CGSS questionnaire to quantify health consumption through the expenditure on social security services. An increase in the acquisition of social security programs by inhabitants correlates with a heightened degree of health consumption, and conversely.

Analysis of empirical results

Comments 1. In your section on heterogeneity analysis, you suggest comparing the findings of other scholars to increase credibility.

Response: Thanks for your useful suggestions. We have added corresponding existing research results to the analysis of income heterogeneity, gender heterogeneity, and heterogeneity in the level of urban development, respectively.

Please see Lines 401-402, 423-424, 443-444 in the revised manuscript.

Conclusion and Policy Implications

Comments 1. The discussion section is too brief and does not go into depth from the findings. It should be integrated with the existing literature and suggest policy implications at the government level.

Response: Thanks for your useful suggestions. We have made relevant revisions to the conclusion section based on the reviewers’ suggestions and have made relevant policy recommendations at the government level based on the conclusion.

Please see Lines 466-468,481-520 in the revised manuscript.

Reviewer #2:

Comments 1. In terms of marginal contribution: Since there has already been research focusing on the impact of low-carbon transition on residents’ health or consumption, the authors should emphasize why it is important to focus on residents’ health consumption within the context of low-carbon transition research. The article should clarify what questions it addresses, what research perspective it adds to the existing literature, its practical significance, and its theoretical implications. Furthermore, the authors mention that this article establishes a theoretical analysis framework to examine the impact of low-carbon transition on residents’ health consumption. However, the theoretical analysis framework presented in the following sections is based on existing literature and does not offer significant theoretical contributions. Therefore, the authors need to reconsider whether the contributions in this section are appropriate.

Response: Thanks for your useful suggestions. We have reworked the theoretical contributions section based on reviewer suggestions.

Please see Lines 71-86 in the revised manuscript.

Comments 2. The logic of the introduction is somewhat tenuous, especially in the second paragraph. The author first mentions the reasons for implementing a low-carbon transition in the second paragraph, then abruptly shifts the topic to the current economic situation and introduces the importance of consumption. It is difficult to emphasize why the low-carbon transition should prioritize the importance of residents’ health.

Response: Thanks for your useful suggestions. We have reworked the second part of the introduction to make it more logical for the article, as suggested by the reviewer.

Please see Lines 56-70 in the revised manuscript.

Comments 3. The micro data of this paper comes from CGSS database in 2018 and 2021. As far as I know, CGSS data are collected in many periods, so why the author only uses the data of these two periods, such as the data of previous years can also be used? If not, it should be specified in the article.

Response: Thanks for your useful suggestions. We have added instructions for using data from these two issues to the article as suggested by the reviewer.

Please see Lines 298-304 in the revised manuscript.

Comments 4. In terms of variable definition, the article defines health consumption based on the purchase of four types of insurance. This method of defining variables is skeptical, and the author should clarify whether any existing literature has adopted this approach. Among the four types of insurance are health insurance and pension insurance; however, the relationship between pension insurance and residents’ health consumption is not closely related. Additionally, the author assigns values based on the number of insurance purchases, which fails to accurately reflect the differences in health consumption among residents. This is because basic health insurance is a social welfare benefit that residents are entitled to. If residents purchase commercial health insurance, their premium expenditures and level of concern for health are often higher, but the variable definition used in the article does not accurately capture this. Existing research on residents’ health consumption typically uses their expenditure on medical and health services as a research variable. If there are relevant variables in the data used in this article, it is recommended that the author include them in the robustness check section to enhance the reliability of the research conclusions.

Response: Thanks for your useful suggestions. Scholars have delineated "health consumption" from various viewpoints, and this paper contends that health consumption encompasses activities aimed at maintaining, enhancing, or restoring health. This includes, but is not confined to, the acquisition of medical services, nutritious food, health insurance, etc., and can be classified into active and passive health consumption. Active health consumption highlights the proactive choices and behaviors of individual consumers, such as investing in health insurance. Conversely, passive health consumption is contingent upon external events, such as the onset of diseases. In contrast to passive health consumption, active health consumption more accurately represents people’ willingness and capacity to engage in health-related expenditures. Furthermore, while pension insurance does not directly offer health services or products, it indirectly promotes an individual’s health by guaranteeing financial stability. A stable income enables individuals to invest more liberally in their health, including access to healthcare services. This paper utilizes the research of Mocan et al. (2003) and integrates the CGSS questionnaire to quantify health consumption through the expenditure on social security services. An increase in the acquisition of social security programs by inhabitants correlates with a heightened degree of health consumption, and conversely.

Furthermore, the database’s exclusion of residents’ healthcare service expenditures precludes the possibility of conducting the essential robustness test. We trust you will comprehend that this paper incorporates the decision to get commercial medical insurance as a robustness test for residents’ health consumption variables.

Please see Lines 324-337, 338-339(Table 3) in the revised manuscript.

Comments 5. In terms of model specification, both Model 1 and Model 2 should account for clustered robust standard errors at the city level to avoid bias in the baseline regression results due to the correlation among individuals within the same city.

Response: Thanks for your useful suggestions. We have modified the treatment of standard errors to enhance their suitability for cluster analysis at the city level.

Please see Lines 257, 321-322(Table 2), 338-339(Table 3), 356-358(Table 4), 390-391(Table 5), 455-456(Table 6) in the revised manuscript.

Comments 6. Mechanism Analysis Section: (1) The definition of the mechanism variable “residents’ health awareness” in this article may not be entirely appropriate. The article is based on the questionnaire questions “In the past four weeks, how often has your work or other daily activities been affected by health problems? Always, often, sometimes = 1; other = 0” to define residents’ health awareness, a potential issue with the questionnaire is that the questions may reflect more on the extent to which residents are affected by health issues in their work and daily lives, rather than their concern for their own health problems. (2) Unlike existing literature that uses urban air quality as a representation of urban environmental quality, this article’s choice of urban green space area as a proxy variable for urban environmental quality may be incomplete. It is recommended that the authors appropriately include air quality for examination in this section.

Response: Thanks for your useful suggestions. (1) Rationale for Assessing Residents’ Health Awareness: This research contends that analyzing the prevalence of health issues encountered by residents in their work or everyday activities can operate as an indirect indicator of their health awareness level. When residents have recurrent health issues that disrupt their employment or everyday activities, such experiences may act as a catalyst for heightened health awareness and encourage individuals to adopt a more proactive approach to health management. Conversely, if inhabitants consider themselves physically strong and well, they may exhibit diminished vigilance and investment in health maintenance, perhaps resulting in a proportionate decline in health awareness. In summary, a probable negative association exists between the prevalence of health issues and the level of individual health awareness; specifically, a growth in health problems is likely inversely connected to the improvement of health awareness. (2) This research primarily posits that urban low-carbon transition influences residents’ health consumption via enhancing the level of habitat environment rather than urban environmental quality, following a literature analysis and reflection. Consequently, according to Zhou’s (2024) research, habitat environment level is assessed by the extent of urban green space. The theoretical analysis and mechanism test have been amended. (References: Zhou, J. Urban ecological human settlements design based on green and low-carbon concept. Journal of Computational Methods in Sciences and Engineering.2024, 24(1), 303-309. DOI: 10.3233/JCM-237049)

Please see Lines 232-247, 376-389, 390-391(Table 5) in the revised manuscript.

Comments 7. Robustness Check Section: (1) Based on the research subjects and scenarios of this paper, using a fixed effects model for estimation is more appropriate. However, the use of a random effects model in this section as a robustness check does not resolve the issues present in the baseline estimation results; rather, the results obtained from the random effects model are coarser than those from the fixed effects model. Additionally, there is a problem in the presentation of the estimation results in this section; according to the estimation results in Table 3, Column 1, it should be a 90% confidence interval, not a 95% confidence interval. (2) The author should not only apply winsorization to income but should also apply it to all continuous variables in the control variables to mitigate the bias caused by outliers on the baseline regression results. (3) The author should consider adding more robustness checks, such as replacing the dependent or independent variables, or changing the level of fixed effects or so on, to enhance the credibility of the baseline regression results.

Response: Thanks for your useful suggestions. (1) As suggested by the reviewers, we removed the tests related to the use of random effects models in the robustness testing section. (2) The correlation test incorporating the substitution of explanatory factors was introduced. Robustness tests were performed to evaluate the decision to obtain commercial health insurance as a measure of home health consumption. Furthermore, we standardize all continuous control variables prior to performing the robustness test.

Please see Lines 324-337, 338-339(Table 3) in the revised manuscript.

Comments 8. Endogeneity Analysis Section: To address potential endogeneity issues in this article, the author uses the urban green space coverage rate from 2006 as an instrumental variable for estimation. However, a possible problem is that this instrumental variable may not satisfy the exclusion restriction assumption, as the urban green space coverage rate can directly affect residents’ health rather than only indirectly influence residents’ health through its impact on urban low-carbon transformation. Therefore, it does not meet the exclusion condition. The author should include relevant tests for the exogeneity and exclusion restriction of the instrumental variable

---

## [Decision Letter · Decision Letter 1]

22 Dec 2024

PONE-D-24-39039R1Does the urban low-carbon transition promote the residents' health consumption?PLOS ONE

Dear Dr. Chen,

Thank you for submitting your manuscript to PLOS ONE. After careful consideration, we feel that it has merit but does not fully meet PLOS ONE’s publication criteria as it currently stands. Therefore, we invite you to submit a revised version of the manuscript that addresses the points raised during the review process.

We look forward to receiving your revised manuscript.

Kind regards,

Najmeh Mozaffaree Pour

Guest Editor

PLOS ONE

Journal Requirements:

Reviewers' comments:

Reviewer's Responses to Questions

Comments to the Author

1. If the authors have adequately addressed your comments raised in a previous round of review and you feel that this manuscript is now acceptable for publication, you may indicate that here to bypass the “Comments to the Author” section, enter your conflict of interest statement in the “Confidential to Editor” section, and submit your "Accept" recommendation.

Reviewer #2: (No Response)

Reviewer #3: All comments have been addressed

2. Is the manuscript technically sound, and do the data support the conclusions?

Reviewer #2: Yes

Reviewer #3: Yes

3. Has the statistical analysis been performed appropriately and rigorously? 

Reviewer #2: Yes

Reviewer #3: Yes

4. Have the authors made all data underlying the findings in their manuscript fully available?

Reviewer #2: Yes

Reviewer #3: Yes

5. Is the manuscript presented in an intelligible fashion and written in standard English?

Reviewer #2: Yes

Reviewer #3: Yes

6. Review Comments to the Author

Reviewer #2: Based on the author’s revisions, the quality of the article has improved to some extent, but the following issues remain:

1. The specific reasons for selecting only the CGSS data from 2018 and 2021 remain unclear. The CGSS data collection spans multiple periods, and the article states that the data from these two years were chosen because low-carbon pilot cities were implemented in 2010, 2012, and 2017. However, the low-carbon transition discussed in this paper uses carbon total factor productivity index as the explanatory variable, which does not conflict with the implementation of low-carbon city pilot policies. I suggest that the author select data from more years or provide a clearer explanation.

2. Regarding the selection of the dependent variable, the article uses the number of participants in pension insurance and health insurance as proxy variables for health consumption. However, the author has not clearly explained the specific reasons for choosing pension insurance as a proxy for health consumption. I feel that the relationship between pension insurance and medical expenses is minimal, and this construction of the dependent variable is inappropriate.

3. In the benchmark regression results of Table 2, column (1) presents the findings using a random effects model. However, based on the research subjects and context of this study, employing a random effects model is not appropriate.

Reviewer #3: The authors have made sufficient revisions to the manuscript, and therefore I recommend that the paper be accepted.

7. PLOS authors have the option to publish the peer review history of their article (what does this mean? ). If published, this will include your full peer review and any attached files.

Do you want your identity to be public for this peer review? For information about this choice, including consent withdrawal, please see our Privacy Policy .

Reviewer #2: No

Reviewer #3: No

---

## [Author Response · Author response to Decision Letter 2]

7 Jan 2025

Comments 1. The specific reasons for selecting only the CGSS data from 2018 and 2021 remain unclear. The CGSS data collection spans multiple periods, and the article states that the data from these two years were chosen because low-carbon pilot cities were implemented in 2010, 2012, and 2017. However, the low-carbon transition discussed in this paper uses carbon total factor productivity index as the explanatory variable, which does not conflict with the implementation of low-carbon city pilot policies. I suggest that the author select data from more years or provide a clearer explanation.

Response: Thanks for your useful suggestions. We have provided a clearer explanation of the data used as suggested by the reviewers.

Please see Lines 294-305 in the revised manuscript.

Comments 2. Regarding the selection of the dependent variable, the article uses the number of participants in pension insurance and health insurance as proxy variables for health consumption. However, the author has not clearly explained the specific reasons for choosing pension insurance as a proxy for health consumption. I feel that the relationship between pension insurance and medical expenses is minimal, and this construction of the dependent variable is inappropriate.

Response: Thanks for your useful suggestions. As suggested by the reviewers，we have modified the measure of population health consumption by applying the CGSS “whether or not commercial health insurance is purchased” to measure residents’ health consumption. We have also revised the other contents of the article.

Please see Lines 267-270, 311(Table 1), 314-323, 324(Table 2), 329-337, 341-342, 346(Table 3), 364 (Table 4), 391-393, 397(Table 5), 462(Table 6) in the revised manuscript.

Comments 3. In the benchmark regression results of Table 2, column (1) presents the findings using a random effects model. However, based on the research subjects and context of this study, employing a random effects model is not appropriate.

Response: Thanks for your useful suggestions. We have removed the test associated with the random effects model as suggested by the reviewer.

---

## [Decision Letter · Decision Letter 2]

20 Jan 2025

PONE-D-24-39039R2Does the urban low-carbon transition promote the residents' health consumption?PLOS ONE

Dear Dr. Chen,

Thank you for submitting your manuscript to PLOS ONE. After careful consideration, we feel that it has merit but does not fully meet PLOS ONE’s publication criteria as it currently stands. Therefore, we invite you to submit a revised version of the manuscript that addresses the points raised during the review process.

Based on the reviewers' feedback, the authors are requested to address the following unresolved issues: (1) Provide a clear and robust justification for the measurement of health consumption indicators, explaining why commercial health insurance is appropriate and considering alternative measures such as health-related expenditures. (2) Revise the selection of control variables by consulting relevant literature and including key factors such as marital status, gender, and family assets. (3) Clarify the sources of endogeneity in the model, provide a detailed rationale for the instrumental variable selection, and include tests for its validity. (4) Reassess the robustness tests, ensuring proper variable assignments and addressing issues like inconsistent treatment of weekly and monthly sports activity measures. These revisions are essential to improve the methodological rigor and credibility of the study.

We look forward to receiving your revised manuscript.

Kind regards,

Najmeh Mozaffaree Pour, PhD

Guest Editor

PLOS ONE

Journal Requirements:

Reviewers' comments:

Reviewer's Responses to Questions

**Comments to the Author**

1. If the authors have adequately addressed your comments raised in a previous round of review and you feel that this manuscript is now acceptable for publication, you may indicate that here to bypass the “Comments to the Author” section, enter your conflict of interest statement in the “Confidential to Editor” section, and submit your "Accept" recommendation.

Reviewer #2: (No Response)

2. Is the manuscript technically sound, and do the data support the conclusions?

Reviewer #2: Yes

3. Has the statistical analysis been performed appropriately and rigorously? 

Reviewer #2: Yes

4. Have the authors made all data underlying the findings in their manuscript fully available?

Reviewer #2: Yes

5. Is the manuscript presented in an intelligible fashion and written in standard English?

Reviewer #2: Yes

6. Review Comments to the Author

Reviewer #2: Comments 1: Regarding the measurement of health consumption indicators. In the revised manuscript, the article uses "whether to purchase commercial health insurance" as a measure of health consumption. It should be noted that why can commercial health insurance effectively measure individual health consumption? Further explanation is needed in the article. In addition, health consumption expenditure can usually be measured using health-related expenditure.

Comments 2: Regarding the selection of control variables. The selection of control variables in the article is relatively arbitrary, and the selection of control variables should strictly refer to relevant literature. The marital status, gender, household registration, family assets, and other variables of the head of the household can all affect the health consumption expenditure of the family, and these variables need to be controlled.

Comments 3: Addressing endogeneity issues. The article uses instrumental variable method to solve the endogeneity problem of the model. Before selecting instrumental variables, it should be clearly explained what the sources of endogeneity in the model are? Additionally, how are instrumental variables measured and what are the basic descriptive results?

Comments 4: Details. For example, on page 16 of robustness test 4.2, the CGSS questionnaire "Do you participate in sports activities during leisure time" was used to measure health consumption, with a value of 3 for "how many times per week" and 3 for "how many times per month". The weekly assignment is the same as the monthly assignment, which is obviously not reasonable. It is recommended to verify.

7. PLOS authors have the option to publish the peer review history of their article (what does this mean? ). If published, this will include your full peer review and any attached files.

**Do you want your identity to be public for this peer review?** For information about this choice, including consent withdrawal, please see our Privacy Policy .

Reviewer #2: No

---

## [Author Response · Author response to Decision Letter 3]

25 Jan 2025

Reviewer #2:

Comments 1. Regarding the measurement of health consumption indicators. In the revised manuscript, the article uses "whether to purchase commercial health insurance" as a measure of health consumption. It should be noted that why can commercial health insurance effectively measure individual health consumption? Further explanation is needed in the article. In addition, health consumption expenditure can usually be measured using health-related expenditure.

Response: Thanks for your useful suggestions. We have added an explanation for the selection of “whether to purchase commercial health insurance” as a measure of residents’ health consumption. In addition, since the questionnaire of the China General Social Survey (CGSS) used in this paper does not contain specific content that directly reflects health-related expenditures, we are unable to use health expenditures as an indicator of health consumption. We hope for your understanding and thank you very much.

Please see Lines 267-282 in the revised manuscript.

Comments 2. Regarding the selection of control variables. The selection of control variables in the article is relatively arbitrary, and the selection of control variables should strictly refer to relevant literature. The marital status, gender, household registration, family assets, and other variables of the head of the household can all affect the health consumption expenditure of the family, and these variables need to be controlled.

Response: Thanks for your useful suggestions. We have added relevant control variables by reviewing existing research results and have revised the full text accordingly.

Please see Lines 300-305, 324(Table 1), 337(Table 2), 358(Table 3), 394(Table 4), 429(Table 5), 494(Table 6) in the revised manuscript.

Comments 3. Addressing endogeneity issues. The article uses instrumental variable method to solve the endogeneity problem of the model. Before selecting instrumental variables, it should be clearly explained what the sources of endogeneity in the model are? Additionally, how are instrumental variables measured and what are the basic descriptive results?

Response: Thanks for your useful suggestions. In this paper, we have elaborated on the sources of endogeneity issues and explained how instrumental variables are measured. Meanwhile, we have further analyzed the statistical significance and economic significance of the regression results of the instrumental variables in depth to ensure the scientific validity and robustness of the findings.

Please see Lines 361-369,380-393 in the revised manuscript.

Comments 4. Details. For example, on page 16 of robustness test 4.2, the CGSS questionnaire "Do you participate in sports activities during leisure time" was used to measure health consumption, with a value of 3 for "how many times per week" and 3 for "how many times per month". The weekly assignment is the same as the monthly assignment, which is obviously not reasonable. It is recommended to verify.

Response: Thanks for your useful suggestions. Upon verification, we have assigned a value of 2 to “how many times per month”.

Please see Lines 347-349 in the revised manuscript.

---

## [Decision Letter · Decision Letter 3]

10 Feb 2025

Does the urban low-carbon transition promote the residents' health consumption?

PONE-D-24-39039R3

Dear Dr. Chen,

We’re pleased to inform you that your manuscript has been judged scientifically suitable for publication and will be formally accepted for publication once it meets all outstanding technical requirements.

Kind regards,

Najmeh Mozaffaree Pour

Guest Editor

PLOS ONE

Additional Editor Comments (optional):

Reviewers' comments:

Reviewer's Responses to Questions

**Comments to the Author**

1. If the authors have adequately addressed your comments raised in a previous round of review and you feel that this manuscript is now acceptable for publication, you may indicate that here to bypass the “Comments to the Author” section, enter your conflict of interest statement in the “Confidential to Editor” section, and submit your "Accept" recommendation.

Reviewer #2: All comments have been addressed

2. Is the manuscript technically sound, and do the data support the conclusions?

Reviewer #2: Yes

3. Has the statistical analysis been performed appropriately and rigorously? 

Reviewer #2: Yes

4. Have the authors made all data underlying the findings in their manuscript fully available?

Reviewer #2: Yes

5. Is the manuscript presented in an intelligible fashion and written in standard English?

Reviewer #2: Yes

6. Review Comments to the Author

Reviewer #2: (No Response)

7. PLOS authors have the option to publish the peer review history of their article (what does this mean? ). If published, this will include your full peer review and any attached files.

**Do you want your identity to be public for this peer review?** For information about this choice, including consent withdrawal, please see our Privacy Policy .

Reviewer #2: No

---

## [Editor Report · Acceptance letter]

PONE-D-24-39039R3

PLOS ONE

Dear Dr. Chen,

I'm pleased to inform you that your manuscript has been deemed suitable for publication in PLOS ONE. Congratulations! Your manuscript is now being handed over to our production team.

Kind regards,

on behalf of

Dr. Najmeh Mozaffaree Pour

Guest Editor

PLOS ONE